# Chemical Constituents from *Albiziae Cortex* and Their Ability to Ameliorate Steatosis and Promote Proliferation and Anti-Oxidation In Vitro

**DOI:** 10.3390/molecules24224041

**Published:** 2019-11-07

**Authors:** Xuelin Shi, Zhongjie Li, Weiwei Cai, Yixiao Liu, Shuangshuang Li, Min Ai, Jiangnan Sun, Bao Hou, Lulu Ni, Liying Qiu

**Affiliations:** 1School of Pharmaceutical Sciences, Jiangnan University, Wuxi 214122, China; sxlgeneral@163.com (X.S.); 6171504016@stu.jiangnan.edu.cn (Z.L.); 2Department of Basic Medicine, Wuxi School of Medicine, Jiangnan University, Wuxi 214122, China; caiweiwei@jiangnan.edu.cn (W.C.); 18861824121@163.com (Y.L.); Liss_@163.com (S.L.); 6182806001@stu.jiangnan.edu.cn (M.A.); 6182806005@stu.jiangnan.edu.cn (J.S.); houbao2015@163.com (B.H.)

**Keywords:** *Albiziae Cortex*, chemical constituents, lignan, steatosis, proliferation, anti-oxidation

## Abstract

This study describes the chemical constituents of *Albiziae Cortex* and their ability to ameliorate steatosis and promote proliferation and anti-oxidation in vitro. Together, five known lignan glycosides, (7S,8R)-erythro-syringylglycerol-β–O-4′-sinapyl ether 9-O-β-D-glucopyranoside (**1**), (+)-lyoniresinol-9′-O-gluco-side (**2**), (−)-lyoniresinol-9′-O-glucoside (**3**), picraquassioside C (**4**), and icariside E5 (**5**), were isolated from the *Albiziae Cortex*. Their structures were elucidated by extensive NMR and high-resolution mass spectrometry analysis and compared with reported data. Oil Red O staining results revealed that compounds **1**, **2**, and **3** attenuated lipid accumulation and lipid metabolic disorders in FFAs (oleate/palmitate, 2:1 ratio, 0.3 mM)-exposed HepG2 cells. The Cell Counting Kit 8 (CCK-8) assay results revealed that compounds **1** and **5** can significantly promote human umbilical vein endothelial cell (HUVEC) proliferation; meanwhile, these compounds did not exhibit significant cytotoxicity against HUVECs. In addition, 2′,7′-dichlorofluorescein diacetate (DCFH-DA) staining results revealed that high glucose (HG)-induced reactive oxygen species (ROS) production was abolished by compounds **1**, **2**, and **3**. This is the first report of the isolation of lignan skeletons from the genus *Albizzia julibrissin* with the ability to ameliorate steatosis and promote proliferation and anti-oxidation activities.

## 1. Introduction

Hepatocyte steatosis is a metabolic syndrome that causes changes in the liver. These changes are characterized by excessive steatosis in hepatocytes and pathological changes including simple fatty liver, fatty liver fibrosis, associated cirrhosis, and liver cancer [1]. It is also a remarkable feature of type 2 diabetes (T2DM), which may lead to cardiovascular disease [2]. Currently, many Chinese herbal medicines have been marketed for the treatment of fatty liver, such as Ginsenosides, Curcumin, and *Lycium barbarum*, which have been reported to prevent fatty liver [3]. Cellular proliferation is not only an indispensable feature of the cell cycle but also the basis for the growth, inheritance, and evolution of organisms. Proliferation plays a key role in physiology and pathology [4]. In many biological processes such as embryogenesis, tissue remodeling, bone development, ovarian circulation, and wound healing, this is a strictly regulated process and a normal process [5]. Oxidative stress is strongly correlated with cell damage, apoptosis, and various cardiovascular complications, with cumulative reactive oxygen species in response to high glucose (HG) being a major cause of cell damage and apoptosis [6]. Considering the pharmacological effects of *Albiziae Cortex*, the aim of this study was therefore to further investigate pharmaceutical ingredients of *Albiziae Cortex* to ameliorate steatosis and antioxidant stress and to promote endothelial cell proliferation activities.

*Albiziae Cortex* is the bark of the leguminous plant *Albizia julibrissin Durazz*, it has been used in traditional Chinese medicine for the treatment of insomnia and swelling [7]. Modern pharmacological studies have demonstrated that the extracts of *Albiziae Cortex* have anti-tumor angiogenesis properties [8]. Our previous studies have shown that the butyl alcohol fraction of the *Albiziae Cortex* can significantly inhibit the activity of fatty acid synthase (FAS). Recently, our research team found that the active constituents of the inhibitory fatty acid synthase in the butyl alcohol part of the *Albiziae Cortex* were mainly concentrated in the 30% ethanol elution section of D101 macroporous resin (as illustrated in Figure 1). Herein, we report five known glycosidic lignan derivatives, (7S,8R)-erythro-syringylglycerol-β–O-4′-sinapyl ether 9-O-β-D-glucopyranoside (**1**) [9], (+)-lyoniresinol-9′-O-gluco-side (**2**) [10], (−)-lyoniresinol-9′-O-glucoside (**3**) [11], picraquassioside C (**4**) [12], and icariside E_5_ (**5**) [13], isolated from the *Albiziae Cortex.* All these compounds were evaluated for their ability to ameliorate steatosis in FFAs (oleate/palmitate, 2:1 ratio, 0.3 mM)-exposed HepG2 cells and their anti-oxidative stress and cell proliferation activity in human umbilical vein endothelial cells (HUVECs).

## 2. Results and Discussion

### 2.1. Compounds Isolated from Albiziae Cortex

Repeated column chromatography of the 30% macroporous resin elution component of the 1-butanol extraction of the ethanolic extract of *Albiziae Cortex* and combined semi-preparative HPLC led to five lignan glycosides. Compound **1** was obtained as a pale-yellow powder and showed negative optical activity ${\left[\alpha \right]}_{D}^{25}$ = −19.0° (c 0.001, MeOH). Its molecular formula was determined as C_28_H_38_O_14_ based on the HR-ESI-MS peak at *m/z* 633.1932 [M + Cl]^−^, indicating 10 degrees of unsaturation. The ^13^C-NMR spectrum showed eight non-protonated carbons and four methyl carbons, indicating the presence of one syringoylglycerol, one glucopyranose moiety, and one sinapyl alcohol (Appendix A). The ^1^H-NMR data gave four methoxy groups at δ H 3.85 (s, 6H) and δ H 3.83 (s, 6H). In addition, the ^1^H-NMR spectrum (Appendix A) showed four aromatic ring protons. Further spectral evidence enabled compound **1** to be identified as (7S,8R)-erythro-syringylglycerol-β–O-4′-sinapyl ether 9-O-β-D-glucopyranoside, which had previously been reported [9].

The known compounds **2**–**5** were identified as (+)-lyoniresinol-9′-O-gluco-side (**2**) [10], (−)-lyoniresinol-9′-O-glucoside (**3**) [11], picraquassioside C (**4**) [12], and icariside E5 (**5**) [13] (Figure 2) by comparison of their NMR and MS data with those reported. All these compounds were first reported from *Albiziae Cortex*.

### 2.2. Effects of Compounds ***1**–**5*** on FFA-Induced Macro-Lipid Droplets and Steatosis

Oil Red O staining results indicated that FFAs (oleate/palmitate, 2:1 ratio, 0.3 mM) significantly induced macro-lipid droplets and steatosis. After treatment with the positive drug (−)-epigallocatechin gallate (EGCG) for 24 h, lipid droplets were significantly reduced (Figure 3). In contrast, the deposition of lipid droplets was slightly alleviated by compound **1** (Figure 3A,B), but when treated with 20 µM, the deposition of lipid droplets was markedly alleviated by compound **1**. As illustrated in Figure 3C,D, treatment with 10 µM compound **2** significantly alleviated the deposition of lipid droplets in FFA-exposed HepG2 cells (Figure 3C,D); this indicated that compound **2** has a good therapeutic effect on FFA-induced lipid metabolism disorder. Compound **3** also markedly alleviated steatosis, and the optimal concentration of compound **3** was 20 µM in this study (Figure 3E,F). Meanwhile, studies have shown that compounds **4** and **5** do not ameliorate steatosis. In this study, lipid accumulation and the protein expression of fatty acid synthase (FAS) and sterol regulatory element-binding protein-1 (SREBP1) [14] were elevated in FFA-exposed HepG2 cells, while treatment with compounds **1**, **2**, and **3** noticeably reversed the effects of FFAs on lipogenesis (vs. FFAs). The results revealed that compounds **1**, **2**, and **3** could be developed as a new drug for the treatment of steatosis.

### 2.3. Effects of Compounds ***1**–**5*** on HUVEC Proliferation

To evaluate the cellular bioactivity in vitro, compounds **1**–**5** were studied on HUVECs at different concentrations: 0 µM, 5 µM, 10 µM, 20 µM, and 40 µM. The results are shown in Figure 4. Compound **1** clearly showed a notable proliferative effect on HUVECs with a maximum effect concentration value is 10 µM in the present study; At an increased dose, compound **5** slightly promoted the proliferation of HUVECs. The CCK-8 (Cell Counting Kit 8) assay results also revealed that the compounds **2**, **3**, and **4** had little effect on cell proliferation viability at concentrations up to 40 µM (Figure 4).

Proliferation of endothelial cells is an important stage in the process of normal life. In view of the above experimental results, five compounds were isolated from *Albiziae Cortex*, and compounds **1** and **5** were found to be effective in promoting the proliferation of HUVECs as well as stimulating angiogenesis [4]. Compound **1** and compound **4** are isomers, but the results showed that their biological activities differ greatly, indicating that the change in the functional group position of the compound is crucial to the biological activity of the compound. In this experiment, we also found that these compounds are not cytotoxic to HUVECs. Because of the very significant biological activity of compound **1**, it may eventually develop into a promising candidate drug for treatment relating to injury repair and related diseases.

### 2.4. Effects of Compounds ***1**–**5*** on HG-Induced Oxidative Stress

2′,7′-dichlorofluorescein diacetate (DCFH-DA) staining results revealed that HG-induced reactive oxygen species (ROS) production was abolished by compounds **1**, **2**, and **3**; when treated with 80 µM, the ROS accumulation was markedly diminished (Figure 5A,B). This experiment also demonstrated that compounds **4** and **5** did not have the pharmacological effect of scavenging ROS. Mitochondrial ROS production has important effects on mitochondrial dynamics and apoptosis; previous studies showed that accumulated reactive oxygen species in response to high glucose are the primary cause of cell damage and apoptosis [15], while anti-oxidant drugs can reduce the damage caused by oxygen free radicals to the blood vessel wall, thereby playing an anti-atherosclerosis role [16]. These results indicate that compounds **1**, **2**, and **3** could considered to be as a new drug for the treatment of atherosclerosis.

## 3. Experimental Section

### 3.1. General

NMR spectra were recorded on a Bruker AV-400 spectrometer (Bruker, Switzerland). ESI-MS spectra were obtained on a Xevo-TQD Ultra-High-Performance Liquid Chromatograph Tandem Quadrupole Mass Spectrometer (MALDI SYNAPT MS, Waters, Dublin, Ireland). Semi-preparative HPLC separation was performed on an LC2488 system equipped with a Waters 1525 pump, UV2998 detector (Waters), Ti-U Nikon inverted microscope (Nikon, Ti-U, Tokyo, Japan), biological safety cabinet (Shandong Brocade Group, Jinan, China), and D101 macroporous adsorption resin (Donghong Chemical Co, Ltd., Jinan, China). Column chromatography was carried out on silica gel (200–300 mesh, Qingdao Marine Chemical Factory, Qingdao, China), Davisil C18 (633N, 50 µm, Waters), and an X-Bridge RP-C18 column (250 × 10 mm, 250 × 4.6 mm, 5 μm, Waters). Optical rotation was determined in MeOH on a Rudolph Autopol IV-T (Rudolph, Hackettstown, NJ, USA) polarimeter.

### 3.2. Reagents

Anhydrous ethanol (AR), methanol (AR, HPLC), dichloromethane (AR), 1-butanol (AR), and ethyl acetate (AR) were purchased from Sinopharm Chemical Reagent Co., Ltd. (Shanghai, China); Dulbecco’s modified Eagle’s medium (DMEM) and fetal bovine serum (FBS) were obtained from Gibco BRL (Carlsbad, CA, USA). Oleate palmitate was acquired from Sigma (St. Louis, MO, USA). Cell counting kit 8 (CCK-8) was purchased from Beyotime Biotechnology Research Institute (Shanghai, China), and Oil Red O reagent was purchased from Nanjing Jiancheng Bioengineering Institute (Nanjing, China). (−)-Epigallocatechin gallate (EGCG) was purchased from BioBioPha Co., Ltd. (Kunming, China). 2‘,7‘-dichlorofluorescein diacetate (DCFH-DA) and N-acetyl-L-cysteine (NAC) were purchased from Beijing Solarbio Science & Beijing Technology Co., Ltd. (Beijing, China).

### 3.3. Material

*Albiziae Cortex* (batch number: 2017110331) was purchased from Anhui Shenghaitang Chinese Herbal Pieces Company and was authenticated by Prof. Jianwei Chen from Nanjing University of Chinese Medicine. HepG2 cells were acquired from the American Type Culture Collection (ATCC, Manassas, VA, USA) and HUVECs were acquired from the Cell Bank of Chinese Academy of Sciences (Shanghai, China).

### 3.4. Extraction and Isolation

The dried *Albiziae Cortex* (20 kg) was crushed, extracted at 80 °C, and refluxed twice with 75% ethanol (100 L) for 2 h each time. The ethanolic extracts were combined and evaporated to dryness under reduced pressure, yielding a yellow crude extract (1.6 kg). The crude extract was suspended in water (2 L) and partitioned successively with ethyl acetate and 1-butanol to yield the two corresponding extracts. The 1-butanol extract (254 g) was fractionated on a macroporous resin column and eluted with deionized water, 30% ethanol, 50% ethanol, 70% ethanol, and 95% ethanol (CH_3_CH_2_OH/H_2_O) to afford four fractions at 30–95%. Fraction 30% was subjected to repeated column chromatography on a silica gel and eluted with a gradient solvent system of CH_2_Cl_2_/MeOH (40:1, 20:1, 16:1, 10:1, 8:1, and 6:1). The 10:1 eluate was then purified by C18 column chromatography with 31%, 42%, and 52% CH_3_OH (H_2_O); fraction 42% CH_3_OH was subjected to semi-preparative HPLC using MeOH/H_2_O as the eluting solvent to give compound **1** (12 mg). Fraction CH_2_Cl_2_/MeOH (8:1) was purified by C18 column chromatography with a gradient solvent system of MeOH/H_2_O (29:71, 36:64, 40:60, and 100:0 for four fractions A–D). Fractions B and C were purified by C18 column chromatography with MeOH/H_2_O and semi-preparative HPLC to give compounds **2** (19.3 mg), **3** (25.6 mg), **4** (21 mg), and **5** (15.2 mg).

### 3.5. Spectral Data

Compound **1** was obtained as a pale yellow powder, showed negative optical activity ${\left[\alpha \right]}_{D}^{25}$ = −19.0° (c 0.001, MeOH), and possessed the molecular formula of C_28_H_38_O_14_ as evidenced by an HR-ESI-MS peak at *m*/*z* 633.1932 [M + Cl]^−^ in combination with its extensive NMR. ^1^H-NMR (400 MHz, CD_3_OD) δ 6.76 (2H,s, H-2′,6′), 6.74 (2H,s, H-2,6), 6.56 (1H,d, *J* = 16.0 Hz, H-7′), 6.34 (1H,dt, *J* = 15.8, 5.5 Hz, H-8′), 4.81 (1H,d, *J* = 7.2 Hz, H-7), 4.65 (4H,s), 4.29 (1H,dd, *J* = 9.0, 4.8 Hz, H-1”), 4.24 (2H,d, *J* = 5.4 Hz, H-9′), 3.96–3.88 (2H,m), 3.85 (6H,s, 3,5-OMe), 3.83 (6H,s, 3′,5′-OMe), 3.77 (1H,s), 3.73–3.63 (4H,m), 3.60 (1H,dd, *J* = 8.4, 3.9 Hz), 3.56–3.39 (4H,m), 3.22 (1H,s). ^13^C-NMR (100MHz, CD_3_OD) δ153.08 (C-3′,5′), 152.39 (C-3,5), 138.12 (C-4′), 134.99 (C-4), 134.13 (C-1′), 133.30 (C-1), 129.95 (C-7′), 128.51 (C-8′), 104.62 (C-2,6), 104.26 (C-1”), 103.51 (C-2′,6′), 85.66 (C-8), 76.94 (C-3”), 76.37 (C-5”), 74.32 (C-2”), 72.64 (C-7), 69.92 (C-4”), 62.16 (C-9), 61.17 (C-9′), 60.23 (C-6”), 55.61 (3′,5′-OMe), 55.28 (3,5-OMe).

Compound **2**, (+)-lyoniresinol-3α-O-β-D-glucopyranoside, was purified as a white powder and showed positive optical activity ${\left[\alpha \right]}_{D}^{25}$ = +36.0° (c 0.001, MeOH). ESI-MS *m*/*z* 617.1981 [M + Cl]^−^, molecular formula of C_28_H_38_O_13_. ^1^H-NMR (400 MHz, CD3OD) δ 6.60 (1H,s, H-8), 6.45 (2H,s, H-2′,6′), 4.44 (1H,d, *J* = 6.2 Hz,H-4), 4.30 (1H,d, *J* = 7.7 Hz, anomeric-H), 3.95–3.81 (6H,m, 5,7-OMe), 3.76 (6H,s, 3′,5′-OMe), 3.67 (2H,dd, *J* = 11.8, 4.9 Hz,H-3a), 3.56 (1H,dd, *J* = 10.9, 6.6 Hz), 3.47 (1H,dd, *J* = 9.8, 3.9 Hz), 3.39 (1H,t, *J* = 7.7 Hz), 3.36 (3H,s), 3.28–3.23 (2H,m, H-2a), 2.77–2.59 (2H,m, H-1), 2.09 (1H,d, *J* = 5.7 Hz,H-3), 1.73 (1H,s,H-2). ^13^C-NMR (100MHz, CD_3_OD) δ 147.59(C-3′,5′), 147.24(C-5), 146.19(C-7), 137.95(C-1′), 137.51(C-6), 133.10(C-4′), 128.81(C-9), 125.03(C-10), 106.47(C-8), 105.55(C-2′,6′), 103.43(C-1”), 76.85(C-5”), 76.54(C-3”), 73.79(C-2”), 70.27(C-4”), 70.11(C-3α), 64.84(C-2α), 61.44(C-6”), 58.80(5-OMe), 55.48(3′,5′-OMe), 55.23(7-OMe), 45.30(C-3), 41.39(C-4), 39.22(C-2), 32.43(C-1).

Compound **3**, (−)-lyoniresinol-3α-O-β-D-glucopyranoside, was obtained as a pale yellow powder, showed negative optical activity ${\left[\alpha \right]}_{D}^{25}$ = −45.0° (c 0.001, MeOH), and possessed the molecular formula of C_28_H_38_O_13_ as evidenced by an ESI-MS peak at *m*/*z* 617.1890 [M + Cl]^−^ in combination with its ^1^H-NMR, ^13^C-NMR, and a comparison with the literature [11]. ^1^H-NMR (400 MHz, CD_3_OD) δ 6.59 (1H,s, H-8), 6.43 (2H,s, H-2′,6′), 4.24 (1H,d, *J* = 5.3 Hz, H-4), 4.15 (1H,d, *J* = 7.7 Hz, H-1”), 3.87 (3H,s, -OCH3), 3.84 (3H,d, *J* = 3.6 Hz, -OCH3), 3.77 (6H,s, 3′,5′-OCH3), 3.72 (1H,d, *J* = 5.2 Hz), 3.69 (1H,d, *J* = 5.1 Hz), 3.62 (5H,dt, *J* = 22.8, 6.5 Hz), 3.49 (1H,dd, *J* = 12.5, 7.2 Hz), 3.44–3.35 (2H,m), 3.30–3.13 (3H,m), 2.70 (2H,t, *J* = 8.0 Hz, H), 2.13 (1H,dd, *J* = 13.1, 6.9 Hz,H-3), 1.70 (1H,d, *J* = 6.3 Hz,H-1). ^13^C-NMR (100 MHz, CD_3_OD) δ 147.61(C-3′,5′), 147.29(C-5), 146.15(C-7), 138.06(C-1′), 137.49(C-6), 133.21(C-4′), 128.83(C-9), 124.84(C-10), 106.41(C-8), 105.72(C-2′,6′), 102.85(C-1”), 76.79(C-5”), 76.58(C-3”), 73.67(C-2”), 70.61(C-3α), 70.16(C-4”), 64.83(C-2α), 61.31(C-6”), 58.72(5-OMe), 55.53(7-OMe), 55.23(3′,5′-OMe), 45.19(C-3), 41.83(C-4), 39.85(C-2), 32.43(C-1).

Compound **4**, picraquassioside C, was obtained as a white powder, showed negative optical activity ${\left[\alpha \right]}_{D}^{25}$ = −24.0° (c 0.001, MeOH), and possessed the molecular formula of C_28_H_38_O_14_ as evidenced by a ESI-MS peak at *m*/*z* 633.1868 [M + Cl]^−^ in combination with its ^1^H-NMR, ^13^C-NMR and comparison with the literature [12]. ^1^H-NMR (400 MHz, CD_3_OD) δ 6.76 (2H,s, H-2,6), 6.74 (2H,s,H-2′,6′), 6.57 (1H,d, *J* = 15.8Hz,H-7′), 6.33 (1H,dt, *J* = 15.8, 5.5 Hz, H-8′), 4.94 (1H,d, *J* = 5.6 Hz,H-7), 4.81 (1H,d, *J* = 7.3 Hz, H-1”), 4.29 (1H,dd, *J* = 9.0, 4.8 Hz, H-8), 4.24 (1H,d, *J* = 5.4 Hz, H-9′), 3.98–3.89 (2H,m), 3.87 (1H,s), 3.85 (6H,s, 3,5-OMe), 3.83 (6H,s, 3′,5′-OMe), 3.77 (1H,s), 3.73–3.58 (4H,m), 3.52–3.41 (4H,m), 3.37 (1H,s), 3.22 (s, 2H). ^13^C-NMR (100 MHz, CD_3_OD) δ 153.09 (C-3′,5′), 152.40 (C-3,5), 138.12 (C-1), 135.02 (C-4′), 134.17 (C-4), 133.31 (C-1′), 129.96 (C-8′), 128.53 (C-7′), 104.66 (C-2,6), 104.29 (C-1”), 103.55 (C-2′,6′), 85.68 (C-8), 76.94 (C-5”), 76.39 (C-3”), 74.34 (C-2”), 72.67 (C-7), 69.95 (C-4”), 62.17 (C-9′), 61.20 (C-6”), 60.25 (C-9), 55.63 (3,5-OMe), 55.31 (3′,5′-OMe).

Compound **5**, icariside E5, was obtained as a white powder, showed negative optical activity ${\left[\alpha \right]}_{D}^{25}$ = −120.0° (c 0.002, MeOH), and possessed the molecular formula of C_26_H_34_O_11_ as evidenced by an ESI-MS peak at *m*/*z* 557.1735 [M + Cl]^−^ in combination with its ^1^H-NMR, ^13^C-NMR, and a comparison with the literature. ^1^H-NMR (400 MHz, CD_3_OD) δ 6.94 (2H, d, *J* = 4.8 Hz, H-2,6), 6.59 (3H, t, *J* = 7.0 Hz, H-2′,5′,6′), 6.50 (1H,d, *J* = 8.0 Hz, H-7), 6.32 (1H, dt, *J* = 15.6, 5.6 Hz, H-8), 4.69 (2H, d, *J* = 7.2 Hz), 4.24 (2H,d, *J* = 5.5 Hz, H-9), 4.03–3.94 (1H, m), 3.92–3.85 (1H,m), 3.84 (3H, s, 3-OMe), 3.82–3.74 (3H, m), 3.72 (1H,s), 3.71 (3H, s, 3′-OMe), 3.67 (1H, d, *J* = 7.2 Hz), 3.62–3.35 (4H, m), 3.19–3.11 (1H, m), 2.99 (1H, dd, *J* = 13.8, 5.5 Hz), 2.74 (1H, dd, *J* = 13.7, 9.4 Hz). ^13^C-NMR (100 MHz, CD_3_OD) δ 152.05 (C-3), 147.02 (C-3′), 143.94 (C-4′), 143.61 (C-4), 137.53 (C-5), 134.01 (C-1), 131.82 (C-1′), 130.10 (C-8), 128.29 (C-7), 121.22 (C-6′), 117.76 (C-6), 114.29 (C-5′), 112.41 (C-2′), 107.75 (C-2), 103.97 (C-1”), 76.67 (C-5”), 76.46 (C-3”), 74.54 (C-2”), 69.85 (C-4”), 65.45 (C-9′), 62.27 (C-9), 61.06 (C-6”), 55.02 (3′-OCH3), 54.89 (5′-OCH3), 41.39 (C-8′), 37.76 (C-7′).

### 3.6. Cell Culture and Treatments

HepG2 cells were cultured in DMEM containing 25 µM glucose, 10% FBS (Fetal Bovine Serum, Gibco, Waltham, MA, USA), 100 U/mL penicillin, and 100 µg/mL streptomycin in a humidified incubator containing 5% CO_2_ at 37 °C. According to the literature reports, exposure of HepG2 cells to FFAs (oleate/palmitate, 2:1 ratio, 0.3 mM) can induce a steatosis model without influencing cell activities [17]. To induce a steatosis model, HepG2 cells were plated at a density of 1 × 10^5^ per well in a 12-well plate and incubated at 37 °C for 12 h. Then, the cells were exposed to FFAs by supplementation with FFAs for 24 h, followed by treatment with compounds **1**–**5** for 24 h (the final concentrations of the compounds were 0, 10, 20, and 40 µM); a positive control group was treated with EGCG for 24 h, and the control groups were cultured in normal DMEM medium. HUVECs were cultured in DMEM medium (low glucose) supplemented with 10% FBS, 100 U/mL penicillin, and 100 µg/mL streptomycin in a humidified incubator containing 5% CO_2_ at 37 °C. The sample compounds were dissolved in DMSO, then further diluted. Cells were plated at a density of 5 × 10^4^ per well in a 12-well plate and incubated at 37 °C for 12 h. The HUVECs were pretreated with compounds **1**, **2**, **3**, **4**, and **5** (80 µM for 12 h) or NAC (500 µM) for 1 h before 35 mM HG incubation for another 24 h.

### 3.7. Oil Red O Staining

Oil Red O staining was used to evaluate lipid accumulation [18]. The HepG2 cells were washed with PBS and subsequently fixed with 4% paraformaldehyde for 30 min. Then, the cells were incubated with Oil Red O reagent (60% Oil Red dye and 40% water) for 20–30 min after washing with PBS and photographed using a light microscope (Nikon, Ti-U, Tokyo, Japan). The average lipid droplet accumulation area was analyzed using Image-Pro Plus 6.0 by using the same parameters.

### 3.8. Cytotoxicity Evaluation and Cell Viability Assay

HUVECs were cultured in DMEM medium (low glucose) supplemented with 10% FBS, 100 U/mL penicillin, and 100 µg/mL streptomycin in a humidified incubator containing 5% CO_2_ at 37 °C. The sample compounds were dissolved in DMSO, then further diluted. Cells were plated at a density of 3 × 10^3^ per well in a 96-well microplate and incubated at 37 °C overnight. The cells were treated with various concentrations of test compounds (the final concentrations of the compounds were 0, 5, 10, 20, and 40 µM) for 48 h. Then, cell viability was determined by CCK-8 assay [19,20].

### 3.9. Intracellular Reactive Oxygen Species Measurement

Intracellular reactive oxygen species (ROS) generation was detected by a fluorescence probe (DCFH-DA) as previously described [21]. The HUVECs were washed three times with PBS and then incubated with 10 µM DCFH-DA for 30 min in the dark at 37 °C. After washing the cells with PBS, photographs were captured using a fluorescence microscope (80i, Nikon). The mean fluorescence intensity was analyzed using Image-Pro Plus 6.0 (Media Cybernetics, Rockville, MD, USA) by using the same parameters.

## 4. Conclusions

Five lignan glycosides were isolated and identified for the first time from *Albiziae Cortex*. The results revealed that three of the isolated compounds (**1**, **2**, and **3**) can markedly alleviate the deposition of lipid droplets and eliminate HG-induced ROS production, while two of the isolated compounds (**1** and **5**) can effectively promote HUVEC proliferation. This is the first report of these lignans from *Albiziae Cortex* and their ability to ameliorate steatosis, promote proliferation, and provide anti-oxidation activities. These compounds exhibited no significant inhibitory activities against HUVECs. This study provides a new basis for further research into new drugs for the treatment of steatosis, atherosclerosis, and trauma.

## Figures and Tables

**Figure 1 molecules-24-04041-f001:**
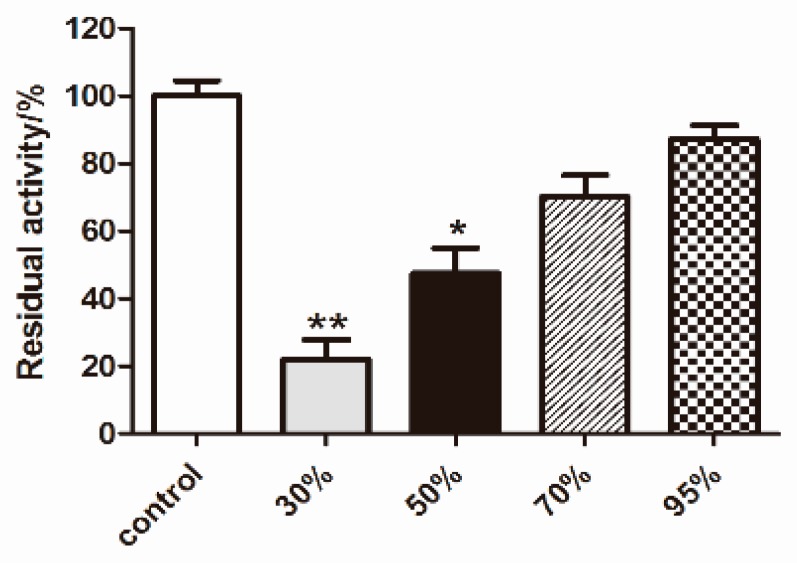
Inhibitory activity of different fractions of macroporous resin on FAS (fatty acid synthase). Different fractions were measured for enzyme activity at the same concentration (5 mg/mL). Values are mean ± SD (*n* = 3), ** *p* < 0.01, * *p* < 0.05 vs. control.

**Figure 2 molecules-24-04041-f002:**
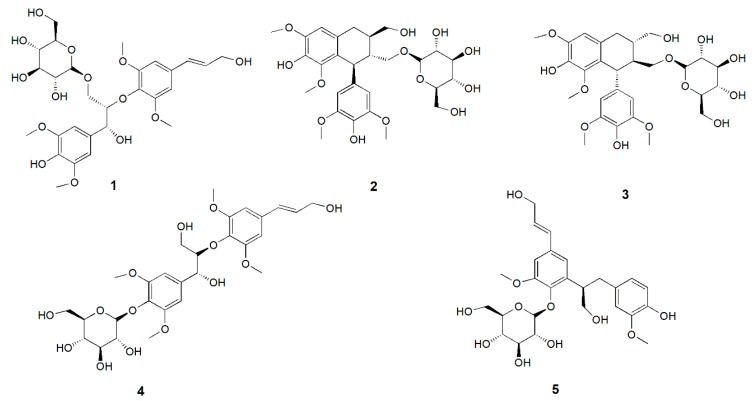
Chemical structures of compounds **1**–**5** identified from *Albiziae Cortex*.

**Figure 3 molecules-24-04041-f003:**
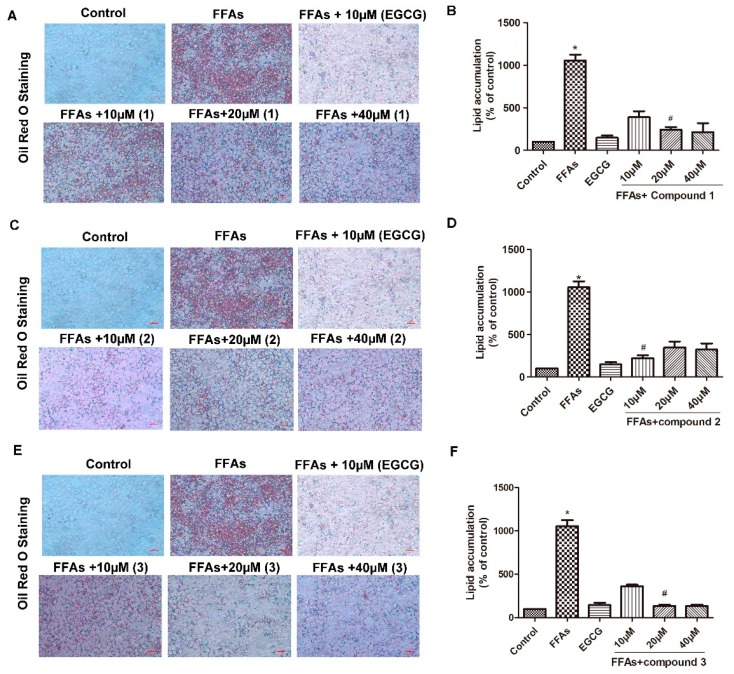
Effect of isolated compounds on steatosis in FFAs-exposed HepG2 cells. The HepG2 cells were treated with FFAs (oleate/palmitate, 2:1 ratio, 0.3 mM) for 24 h, then treated with compounds **1**–**3** for 24 h. (**A**) Compound **1** reversed the effect of FFAs on lipogenesis; (**B**) Lipid accumulation area values. (**C**) Compound **2** reversed the effect of FFAs on lipogenesis; (**D**) Lipid accumulation area values. (**E**) Compound **3** reversed the effect of FFAs on lipogenesis; (**F**) Lipid accumulation area values. (−)-Epigallocatechin gallate (EGCG) is a positive control group; Oil Red O staining showed lipid accumulation, original magnification ×400; Values are mean ± SD (*n* = 3). * *p* < 0.05 vs. control, ^#^
*p* < 0.05 vs. FFAs.

**Figure 4 molecules-24-04041-f004:**
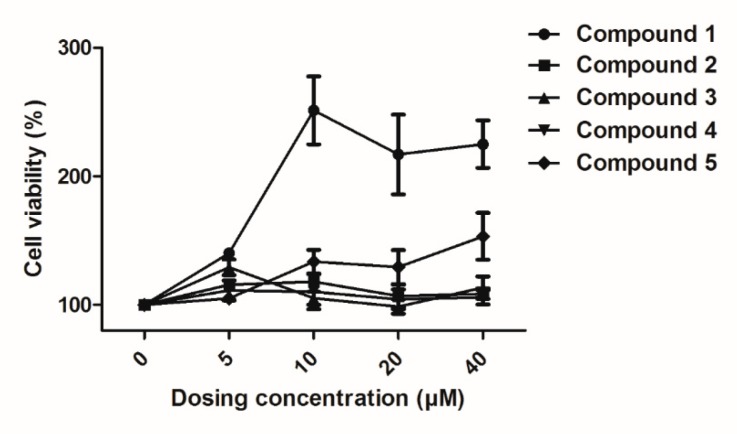
Effects of compounds **1**–**5** on the proliferation of human umbilical vein endothelial cells (HUVECs). HUVECs were cultured with different concentrations (0–40 µM) of compounds. Cellular proliferation was assessed using the Cell Counting Kit 8 (CCK-8) assay after 48 h. Data are expressed as the mean ± SEM (*n* = 3) of three individual experiments.

**Figure 5 molecules-24-04041-f005:**
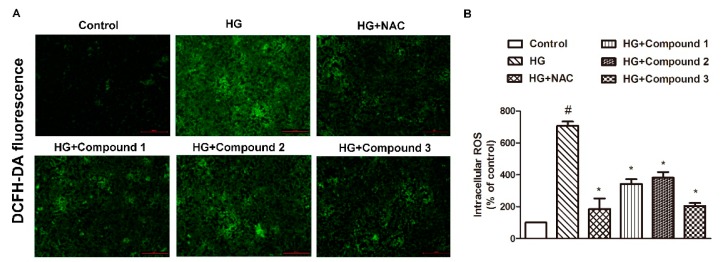
(**A**) Compounds **1**, **2**, and **3** inhibited high glucose (HG)-induced reactive oxygen species (ROS) production. The HUVECs were pretreated with **1**, **2**, and **3** (80 µM) for 12 h or with NAC (N-acetyl-L-cysteine, 500 µM) for 1 h before 35 mM HG incubation for another 24 h. (**A**) Intracellular levels of ROS were detected by 2′,7′-dichlorofluorescein diacetate (DCFH-DA) fluorescence (×200). (**B**) Intracellular ROS fluorescence values. The NAC group is a positive control group. Values are mean ± SD. ^#^
*p* < 0.05 vs. control, * *p* < 0.05 vs. HG, *n* = 3/group.

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
