# Peer review of "Chemical Constituents from Albiziae Cortex and Their Ability to Ameliorate Steatosis and Promote Proliferation and Anti-Oxidation In Vitro"

_molecules, 2019, doi:10.3390/molecules24224041_

Round 1

Reviewer 1 Report

The manuscript describes the isolation from Albiziae Cortex and biological evaluation of five known lignan glycosides. The authors found that three of the compounds could markedly alleviate the deposition of lipid droplets and eliminate HG-induced ROS production, thus showing anti-steatosis and anti-oxidant activity. Moreover, two compounds were found to promote the proliferation of human umbilical vein endothelial cells. The work is well organized, but several grammar mistakes are spread throughout the manuscript; also, several acronyms are not clearly explained not even in brackets. However, my major concern regards the absolute configurations of the compounds, which are indicated although no proof is provided. In my opinion, it should be necessary having at least optical rotation measurements when dealing with known compounds. Otherwise, it is necessary to establish the absolute configuration experimentally. I suggest to carry out these tests and make appropriate corrections to the language of the manuscript before publication on Molecules. 

Author Response

Thank you for your suggestions on this paper, we made appropriate modifications to the grammatical errors that occurred, and added explanations for several acronyms in brackets. Similarly, according to the recommendations, we also measured the optical rotation of compound 1-5, and finally calculated the specific rotation value of the compounds. Please refer to the manuscript for the measurement results. Thank you!

Reviewer 2 Report

the amount of compound extracted is very low for example compound 1 (12 mg). 

this amount of each compound was enough to perform all the experiment, it is doubtful.

Author Response

Thank you very much for reviewing this paper. The 12 mg given in the article is the first compound 1 isolated by semi-preparative HPLC. After NMR identification, we recycle it for pharmacological experiments. Of course, we can still separate the remaining samples by semi-preparative HPLC to isolate our target product. Thank you!

Reviewer 3 Report

I think that this is a paper properly designed to demonstrate the benefits of Albizia bark to improve the symptoms of stenosis. It is not clear to me if the bark extract comes from a single species, Albizia julibrissin (in italics please) Duzazz., Or from a mixture of species of the genus. This work contributes to give scientific explanation about one of the elements based on traditional medicine, which in Europe are less known.

Author Response

 Thank you very much for reviewing this paper, the bark extract comes from a single species of Albiziae Cortex, not from a mixture of species of the genus, the manuscript refers to the Albiziae Cortex belonging to the genus Albizzia julibrissin. Albiziae Cortex is the bark of the leguminous plant Albizia julibrissin Durazz, it is a traditional Chinese medicine. Related vocabulary appearing in the article has been modified to italic. Thank you!

Round 2

Reviewer 1 Report

The revisons requested have been made. I suggest publishing the manuscript in the current version.

Reviewer 2 Report

The manuscript was  reviewed